# The Extracellular Matrix Stiffening: A Trigger of Prostate Cancer Progression and Castration Resistance?

**DOI:** 10.3390/cancers14122887

**Published:** 2022-06-11

**Authors:** Carole Luthold, Tarek Hallal, David P. Labbé, François Bordeleau

**Affiliations:** 1Centre de Recherche sur le Cancer, Université Laval, Québec, QC G1R 3S3, Canada; carole.luthold@gmail.com; 2Division of Oncology, Centre de Recherche du CHU de Québec-Université Laval, Hôtel-Dieu de Québec, Québec, QC G1R 3S3, Canada; 3Cancer Research Program, Research Institute of the McGill University Health Centre, Montréal, QC H4A 3J1, Canada; tarek.hallal@mail.mcgill.ca; 4Division of Urology, Department of Surgery, McGill University, Montréal, QC H4A 3J1, Canada; 5Département de Biologie Moléculaire, Biochimie Médicale et Pathologie, Faculté de Médecine, Université Laval, Québec, QC G1V 0A6, Canada

**Keywords:** prostate cancer, extracellular matrix stiffening, androgen receptor, androgen deprivation therapy, mechanosensing, metastasis

## Abstract

**Simple Summary:**

Prostate cancer is a malignancy that affects a high number of men all over the world. When indolent, prostate cancer can remain silent for years without needing medical intervention. However, aggressive prostate cancer can grow fast, resist treatment, and cause morbidity and ultimately death. Uncovering mechanisms of prostate cancer disease progression and therapy resistance is important to develop new treatments that help patients live longer and healthier. The extracellular matrix, which provides physical support for tissues and organs, is emerging as an important mediator of disease, especially in cancer. In this review, we examine how extracellular matrix alteration, primarily through stiffening, can affect prostate cancer disease course. We look at mechanisms that involve the androgen receptor, which lies at the center of the disease transcriptional landscape, as well as alternative pathways that are androgen receptor-independent.

**Abstract:**

Despite advancements made in diagnosis and treatment, prostate cancer remains the second most diagnosed cancer among men worldwide in 2020, and the first in North America and Europe. Patients with localized disease usually respond well to first-line treatments, however, up to 30% develop castration-resistant prostate cancer (CRPC), which is often metastatic, making this stage of the disease incurable and ultimately fatal. Over the last years, interest has grown into the extracellular matrix (ECM) stiffening as an important mediator of diseases, including cancers. While this process is increasingly well-characterized in breast cancer, a similar in-depth look at ECM stiffening remains lacking for prostate cancer. In this review, we scrutinize the current state of literature regarding ECM stiffening in prostate cancer and its potential association with disease progression and castration resistance.

## 1. Introduction

In 2020, prostate cancer was the second most diagnosed cancer in men globally, and the fifth cause of cancer-related mortality [1]. The androgen receptor (AR), a ligand-inducible transcription factor, is key in controlling not only normal prostate homeostasis, such as cell proliferation and differentiation [2], but also prostate cancer initiation, growth, and progression. Since the AR is activated by androgens, androgen deprivation therapy (ADT), which can be achieved either through surgical castration (orchiectomy) or chemical castration (using Gonadotropin-Releasing Hormone agonists or antagonists), is the mainstay treatment line for prostate cancer [3]. Unfortunately, despite a robust initial response, between 20% and 30% of cases progress towards an aggressive state of the disease, referred to as “castration-resistant prostate cancer”, or CRPC [4]. The vast majority of CRPC patients develop metastatic disease (mCRPC) and have a poor prognosis with a median overall survival of 18 months [5]. Essentially no patients with mCRPC are cured of their disease. Despite very low androgen levels under ADT, AR-mediated signaling remains an important driver of CRPC progression due to AR transactivation through several mechanisms, including AR amplification to increase sensitivity to androgens, deregulation of coactivators/corepressors, as well as androgen-independent AR activation through the epidermal growth factor receptor (EGFR)/mitogen-activated protein kinase (MAPK) and phosphoinositide 3-kinase (PI3K)/AKT signaling pathways [6]. However, the prostate cancer progression molecular landscape is not solely defined by androgens and the AR, especially during later stages of the disease, as several other mechanisms come into play, such as WNT- and fibroblast growth factor (FGF)-related pathways [7,8].

Over the last three decades, evidence about the crucial role of the extracellular matrix (ECM) in cancer has exploded [9]. More recently, the matrix has emerged as a predictive and a diagnostic tool, as well as a novel therapeutic target for cancer treatment [9]. While it is already known that ECM alterations are crucial drivers of cancer progression towards metastasis [9], we are only beginning to expose the mechanisms underlying the cell mechano-adaptative response. Better understanding of these mechanisms is critical to improve patient care and disease outcomes. Particularly, ECM dysregulation has been extensively studied in breast cancer, and several ongoing clinical trials are targeting the ECM or the cell responses associated with these deregulations [10,11]. However, research is still catching up regarding prostate cancer.

In this review, we aim to dissect the current knowledge regarding ECM alterations, more specifically ECM stiffening, in prostate cancer disease and propose potential links of such events with disease progression and castration resistance.

## 2. The Extracellular Matrix Stiffness Contributes to the Progression of Prostate Cancer

### 2.1. Overview of the Extracellular Matrix

The ECM is a complex and dynamic interconnected network of macromolecules that surround cells and provide a supporting scaffold to maintain tissue structure and homeostasis as well as acting as a critical driver of morphogenesis. For example, ECM fibers in the mammary gland accumulate around the duct and are axially oriented prior to branching morphogenesis [12,13]. The ECM architecture and composition confer elasticity and strength to a tissue according to its specific needs and functions [14,15,16].

The most abundant structural component of the ECM is collagen, which accounts for around 30% of the total protein present in the body [17,18]. There are different types of collagen, classified according to their assembly as either fibrillar (collagens I, II, III, V, XI, XXIV, and XXVII) or non-fibrillar scaffolds [17,19]. Fibrillar collagens, mostly type I collagen, provide mechanical strength to the ECM, enabling resistance to deformation and rupture [17,18,20]. Nevertheless, the biomechanical properties of the ECM depend on the specific composition and local concentrations of matrix constituents as well as their precise organization and orientation, creating a highly organized topology that contributes to the functional properties of the matrix [21,22]. Notably, other ECM proteins, including the glycoproteins laminins, fibronectins, and tenascins, help to maintain a cohesive network [9,14]. The overall ratio between the different ECM components, as well as their organization, further influence how cells sense the mechanical properties of a tissue [23,24]. Additionally, cell-directed alignment and organization of matrix components, e.g., collagen and fibronectin, contribute significantly to the matrix stiffness compared to the passive mechanical properties of the matrix alone [25,26]. Additionally, once assembled in the extracellular space, collagen fibers are strengthened by covalent crosslinks within and between the constituent collagen molecules. For instance, the lysyl oxidase (LOX) family of extracellular enzymes can covalently crosslink collagen fibers, and thus increase the stiffness of the ECM [27]. Together, these modifications occurring in the matrix orchestrate and fine-tune its remodeling, which is a crucial process for normal development and organ homeostasis.

### 2.2. Cancer, Matrix Remodeling, and Stiffness

Pathological conditions can promote aberrant and extensive ECM deposition and remodeling, acting as key players driving disease progression [28,29]. Tumors have an inherently altered ECM that contributes to the progression of the disease [27,30]. Notably, the ECM is one of the first elements of the microenvironment shown to be altered in tumors relative to normal tissue, and ECM alteration is now considered a hallmark of solid tumors [31,32]. In fact, high ECM density in mammary tissues is considered an important risk factor for tumor initiation and metastasis [33,34]. Cancer cells recruit host stromal cells, such as cancer-associated fibroblasts or tumor-associated macrophages, that work in concert to cooperatively remodel the ECM through several mechanisms [34,35,36]. These cells produce and release different ECM components and remodeling enzymes, but also reorganize and degrade the existing ECM [9]. Together, these modifications result in a pervasive dense fibrous tissue that typically surrounds the tumor and contributes to the increased local tissue stiffness observed within tumors. In fact, this remodeling is constantly active and the ECM within a tumor gradually stiffens over time [9,37,38].

One of the principal mechanisms that drive tumor stiffening is the excessive deposition of ECM components, most notably of type I collagen [9,37,39]. Notably, cells also remodel the ECM through their contractile abilities and can reorient and linearize the collagen fibers during malignant transformation and metastatic dissemination [38,40]. In the mouse mammary gland, local cell invasion is predominantly oriented along aligned collagen fibers, suggesting that radial alignment of collagen fibers relative to tumors facilitates invasion [40]. In vitro, it has been shown that primary tumor explants cultivated in a randomly organized collagen matrix are able to align collagen fibers, resulting in the individual tumor cell migration along radially aligned fibers [40]. The cell’s ECM remodeling abilities are also facilitated by surface expression or secretion of specialized proteases, including matrix metalloproteinase proteins (MMP), a disintegrin and metalloproteinases (ADAMs), and ADAMs with thombospondin-1 motifs (ADAMTS), that can cleave and degrade ECM components [41]. In addition, ECM stiffening is also linked to increased LOX activation or protein expression level [38,42]. High LOX activity in various cancers contributes to increased ECM stiffness, and LOX overexpression promotes metastasis [38,42,43]. Moreover, increased LOX expression is associated with an alignment of fibrillar collagens and tissue stiffness and promotes the growth and invasion of pre-malignant tissue [38]. Altogether, changes of the biochemical composition and biomechanical state of the ECM create proper conditions that facilitate tumor emergence as well as cancer cell invasion and treatment resistance. Particularly, growing evidence indicates that ECM stiffening is a critical driver of tumor growth, invasion, and metastasis in prostate cancer [44,45].

Overall, cancerous lesions tend to be stiffer (or feel harder upon palpation) than benign tissue [45,46,47]. This differentially stiffened tissue mass allows for tumor detection through physical palpation. For instance, digital rectal exam is a key diagnostic tool—directly linked to tumor stiffness—for prostate cancer detection [48,49]. A stiffer microenvironment promotes tumor initiation [33], invasion, and metastasis [38,50,51,52,53,54,55], enhances immune cell infiltration [56], facilitates the epithelial–mesenchymal transition through TGF-β [57], promotes stem cell differentiation [58], alters growth factor secretion and signaling, and increases angiogenesis and vessel permeability [59]. Accordingly, matrix fibrosis and increased stiffness have become a diagnostic marker and an indicator of poor prognosis in several cancers, most notably for lung, breast, and liver cancers [9,37,38,60,61,62,63,64].

### 2.3. Evolution of the ECM during Prostate Cancer Development, Progression, and Invasion

From early stages to metastatic disease, growing evidence highlights an apparent relationship between prostate tumors, ECM component changes, and tissue stiffness. Prostate tissue stiffness has been measured on the entire prostate or on biopsies from men with suspected prostate cancers. As observed in tumors from other organs, these measurements have revealed that malignant prostate tissues are almost 60% stiffer than benign prostate tissues [65,66,67]. In addition, magnetic resonance elastography, a non-invasive method to quantify tissue mechanical properties, has been suggested as an effective way to predict lymph node metastasis in prostate cancer patients [68]. At a cellular level, stiff matrices were shown to induce a phenotypic switch in metastatic cancer cells, which was accompanied by an increase in drug resistance. More specifically, when PC-3 cells were grown on stiffer matrices, cells displayed a reversible phenotypic plasticity, characterized by altered gene expression and changes in cellular morphology. This cellular response was accompanied by decreased sensitivity to paclitaxel, a chemotherapeutic agent used in the treatment of several solid tumors, including the prostate [69]. These results highlight the role of mechanical cues not only in disease progression, but also in response to treatment [69].

In agreement with the tumor tissue stiffening, an increase of type I collagen synthesis was observed in activated peri-acinar fibroblasts adjacent to prostatic intraepithelial neoplasia, which is considered as a precursor lesion of prostatic adenocarcinoma [70,71]. This suggests that the increase in collagen content within the prostate tissue is associated with early development steps of prostate cancer (illustrated in Figure 1). Patients’ data also show a strong positive correlation between collagen-mediated tumor stiffness and tumor evolution and aggressiveness. Recently, the quantification of ECM elements in prostate adenocarcinoma of different Gleason scores revealed an increase of collagen fibers in the tumor area compared to the non-tumor area [72]. Gleason score is the standard grading system used to assess the disease’s aggressiveness [73,74]. First, Gleason grades are determined mainly based on the visual assessment of prostate cells and glandular morphology on hematoxylin/eosin-stained biopsy sections by a pathologist, while the Gleason score is calculated from the addition of the two predominant Gleason grades in the tumor [73,74]. As a result, the correlation between higher collagen density and higher Gleason score suggests that ECM stiffening is associated with more aggressive prostate cancer. Additionally, a reorganization of collagen fibers is observed during prostate cancer progression and correlates with higher Gleason scores [66,75,76]. A distinct pattern of collagen distribution exists for each Gleason score. In a large set of biopsy tissues, collagen structures were found to be more aligned in malignant cores [66,76]. In addition, the increased stiffness is correlated with an increased alignment of collagen fibers and with a higher Gleason score in malignant prostatic lesions [65,66,67]. In summary, collagen fibers tend to be more oriented, and the matrix stiffer, as prostate cancer becomes more aggressive (Figure 1).

While collagen deposition is a major player of tissue stiffening and disease progression, other ECM components have been identified as important actors in disease progression, the emergence of metastatic prostate cancer, and with treatment resistance. For instance, the proteoglycan versican has been proposed as a prognostic factor in prostate cancer. Notably, versican is overexpressed and accumulates in the peritumoral stroma of prostate cancer [77]. Using the human prostate cancer cell lines LNCaP, PC-3, and DU145, it has been shown that the formation of a versican-rich pericellular matrix enhanced prostate cancer cell motility and could contribute to the development of locally invasive disease [78]. Furthermore, versican expression increases with the acquisition of docetaxel resistance in PC-3 cells, a chemotherapeutic agent used for prostate cancer, suggesting a role of this ECM protein in resistance to treatments [79]. Accordingly, a low versican concentration in the peritumoral stroma of patients is associated with a significantly improved progression-free survival compared to patients with high levels of versican [80]. Cancer-associated fibroblasts also produce a fibronectin-rich ECM with an anisotropic orientation of fibers, which guides cancer cells to migrate directionally [81]. Cancer-associated fibroblasts align the fibronectin matrix by increasing non-muscle myosin II-mediated contractility and traction forces [81]. It has also been shown that the expression of the glycoprotein tenascin-C is increased in the stromal microenvironment in human prostate cancer [71,75,82]. Notably, tenascin-C overexpression was significantly correlated with a lower overall survival [82]. High levels of tenascin-C expression in prostate cancer stroma were also significantly associated with lymph node metastasis and clinical stage [82]. Overall, the increased synthesis of tenascin-C predicts a poor prognosis in prostate cancer [82]. Finally, the matricellular protein osteopontin was identified as part of an ECM signature in both mCRPC and bone metastasis [83]. Notably, both the protein and mRNA expression levels of osteopontin were upregulated in mCRPC compared to hormone-sensitive prostate cancers in organoid models [83].

To sum up, modifications of prostatic matrix elements strongly correlate with greater Gleason scores and can contribute to predicting the pathological staging of prostate cancer. Moreover, the ECM stiffening and its constituents emerge as prognostic biomarkers of prostate cancer progression and metastasis. For instance, in metastatic colon cancer, tissue stiffness was shown to support liver metastasis [84]. Among the preferred sites of prostate cancer metastases, the bone has the lion’s share (84%), followed by distant lymph nodes (10.6%), the liver (10.2%), the thorax (includes lung, pleura, mediastinum, and other respiratory organs, 9.1%), and the brain (3.1%) [85] (Figure 1). This bone tropism, which predominantly targets the spine [86], remains poorly understood at the mechanistic level. Bone metastasis affects the quality of life of patients with advanced disease by inflicting several skeletal morbidities which could manifest as pain, fractures, hypercalcemia, and spinal cord compression. Therefore, elucidating mechanisms of prostate cancer metastasis is critical, and ECM stiffening and activation of related signaling pathways are compelling avenues of investigation. Along these lines, there is substantial evidence indicating that critical cellular pathways involved in CRPC disease, such as PI3K/Akt and MAPK [87,88,89,90], are activated in response to ECM stiffening. In fact, there are multiple similarities between signaling pathways’ activation resulting from ECM stiffening and both AR-dependent and independent resistance mechanisms in CRPC and investigating this interplay will likely provide a better understanding of CRPC.

### 2.4. ECM Biochemical and Mechanical Sensing by Integrin-Mediated Adhesion Complexes and the Consequence for Cellular Behavior

At the molecular level, the cells–ECM interaction is mediated by various adhesion receptors, most notably by the integrins, that allow direct connection between the cell cytoskeleton and the surrounding matrix [15,16]. Integrins always come as heterodimeric pairs consisting of an α and β subunit (illustrated in Figure 2). There are 18 α and 8 β subunits which can combine in at least 24 distinct integrin pairs [91]. Each integrin pair exhibits a specific binding affinity to different ECM ligands, including collagens and fibronectin [91]. For instance, the α5β1 integrin pair has more affinity for fibronectin, while the α2β1 pair has more affinity for type I collagen [91]. This confers cells an ability to recognize and bind to specific ECM components depending on their integrin expression patterns [91,92].

Once in contact with its ligand, integrins recruit large and dynamic complexes of proteins, including actin-binding, adaptor, and signaling proteins, such as talin, focal adhesion kinase (FAK), and members of the Src family (Figure 2A) [93,94]. Together, clustered integrins and their associated proteins form focal adhesion complexes [94,95,96]. Focal adhesion formation initiates many signaling cascades’ activation, including the MAPK and PI3K/Akt pathways, which in turn modulate cytoskeletal assembly, allowing the regulation of cellular functions such as proliferation, morphology, and motility [95,97]. However, integrins are also the front-line sensors of mechanical cues from the ECM [98]. As such, integrins are continuously submitted to forces transmitted between cells and the ECM that affect their properties, including ligand-binding kinetics, conformation, activation, and clustering (reviewed in [98]).

In addition to integrins, cells possess “molecular devices” that are defined as cellular mechanosensors [28]. Their mechanosensory functions are conferred by force-induced status changes, including post-translational modifications, intracellular shuttling, as well as protein stretching/unfolding and modulation of protein–protein interactions [28,99]. FAK represents an excellent example since intracellular forces induce conformational change that allows its autophosphorylation [100,101]. Another good example is the actin-binding protein filamin. In response to mechanical force, filamins undergo conformational changes and unfolding events that change their affinity for binding partners and expose cryptic binding sites, leading to recruitment of additional components [102,103,104,105]. Finally, the transcriptional coactivator Yes-associated protein 1 (YAP1) is a well-established mechanical sensor since a stiffer matrix induces its nuclear translocation (Figure 2A) [106,107]. Overall, these changes provide a mechanism by which cellular mechanosensors convert mechanical cues into signal transduction pathways that regulate gene expression or cytoskeleton dynamics, and ultimately affect cellular behavior [28,108,109].

ECM stiffness is, however, a passive mechanical property. Thus, cells must rely on actomyosin contractility to generate the forces required to activate the mechanosensing pathways (reviewed in [110,111,112]). In addition, several signaling loops ensure that cells maintain a balance between their contractility levels and the mechanical properties of the underlying ECM, thus maintaining a mechanical homeostasis [30,111]. It is therefore not surprising that mechanosensing pathways are overactivated in stiff, aggressive tumors [113,114,115]. However, tumor cells often display altered mechanical properties, most notably characterized by increased contractility levels which scale with their metastatic potential [116,117] (reviewed in [118]). Accordingly, the metastatic PC-3 prostate cancer cell line is more contractile than the PrEC primary prostate cell line [117]. Moreover, this also means that mechanosensing pathways have a lower stiffness activation threshold in tumor cells, implying that they would be more responsive to changes in the ECM stiffness. Consequently, signaling crosstalk downstream of integrin activation, such as the MAPK pathways, can also be upregulated in more contractile cells [116]. Together with the increased tumor stiffness, this contractile phenotype creates a powerful feedforward loop that is thought to exacerbate disease progression and could influence the therapeutic response.

## 3. ECM Stiffness and Mechanisms Promoting Resistance to AR-Targeted Therapies

Since prostate cancer maintains its reliance on androgen signaling to thrive, ADT remains the gold standard for treating advanced prostate cancer. In addition, once prostate cancer progresses to become resistant to castration, the next line of therapy typically still involves targeting AR signaling through the use of AR signaling inhibitors (ARSI) such as enzalutamide, which inhibits androgen binding to the AR, AR nuclear translocation, and its binding to *cis*-regulatory elements [119,120]. Unfortunately, despite these therapeutic efforts, prostate cancer invariably progresses towards drug resistance [121,122,123]. This occurs through reactivation of AR signaling or the emergence of an AR-negative cancer that no longer relies on AR signaling, such as in neuroendocrine CRPC [124]. However, the exact mechanisms underlying the transition from androgen-dependent prostate cancers to CRPC remain incompletely understood. Interestingly, increased ECM stiffness seems closely correlated with CRPC. Indeed, numerous evidence suggests a connection between cells’ response to matrix stiffness and mechanisms underlying AR-dependent and independent ADT bypass.

### 3.1. AR-Centric Mechanisms

The mechanisms that allow AR reactivation in the absence of androgens in CRPC are multiple and complex. These include AR amplification and overexpression, AR mutations, expression of constitutively active AR variants, mutations or dysregulation of coactivators/corepressors, and promiscuous androgen-independent AR activation by other factors. Here, we provide an overview of AR-dependent molecular mechanisms which drive CRPC and their potential crosstalk with ECM stiffening in promoting castrate-resistant growth of prostate cancer.

#### 3.1.1. AR Overexpression and Expression of Constitutively Active AR Variants under the Control of ECM Stiffening

Increased AR expression is often observed in CRPC [125,126,127]. While AR overexpression can be linked to AR gene amplification [126], it is tempting to speculate that matrix stiffening could also impact AR expression. Indeed, more than 1500 genes have been shown with an altered expression in human mammary epithelial cells in response to perturbations in matrix stiffness [128]. Thereby, it would be relevant to determine if ECM stiffening could promote AR overexpression in CRPC.

Several AR alternative splicing isoforms have been detected in clinical samples, within the normal prostate, in primary prostate tumors, as well as in metastases [129,130,131,132,133,134,135,136]. However, the greatest levels of AR alternative splicing isoforms were observed in CRPC [130,132,133,135,136]. Several AR splice variants lack the ligand-binding domain but retain their ability to bind DNA in the absence of androgens, thus displaying constitutive activity [137]. For instance, AR-V7 (also termed AR3) and AR-V3 have been described as constitutively active and are recurrently expressed in CRPC [129,130,136,138,139]. Furthermore, AR-V7 has been strongly associated with ARSI resistance (e.g., enzalutamide and abiraterone acetate), tumor growth, and poor patient prognosis [133,135,140,141]. Consequently, AR splice variants are central players in CRPC. Of note, we have shown that a stiffer matrix regulates the splicing events of fibronectin splice variants in both in vitro and in vivo mammary tumors through the activation of PI3K/Akt signaling [142]. Furthermore, the deregulation of the splicing machinery in prostate cancers is highly correlated with the Gleason score as well as with AR-V7 expression levels [143]. Together, these results lead to a fascinating possibility, where expression of AR splice variants could be regulated through the ECM stiffening.

#### 3.1.2. Dysregulation of AR Cofactors: Filamin and YAP as Mechanosensory Coactivator Factors of AR-Mediated Transcription in Response to a Stiff Matrix in CRPC

In the canonical model, androgens (e.g., testosterone, dihydrotestosterone) binding to the AR induces its dimerization, followed by its nuclear translocation, where it binds the androgen response element to control transcription [144]. AR-mediated transcriptional activity is modulated by multiple AR coactivators [55,56,57]. Interestingly, the mechanosensory protein filamin interacts with the AR and has been identified as a positive modulator of AR nuclear translocation and activity [145]. Since filamin stretching in response to matrix stiffening modulates molecular interactions, it is tempting to speculate that such stretching could induce an increased interaction with AR, consequently enhancing AR transcriptional activity. Such process could facilitate resistance to ADT by allowing AR-signaling activation despite low androgen levels.

Another mechanosensory protein has been identified as a binding partner and positive regulator of AR nuclear translocation and activity: the transcriptional coactivator YAP [146,147]. Data from prostate cancer cell lines and patient tissues revealed that YAP and AR form a protein complex that primarily occurs in the cell nucleus [146]. Interestingly, both the complex formation and its nuclear localization are androgen-dependent in castration-sensitive LNCaP cells, but androgen-independent in castration-resistant C4-2 cells [146]. Furthermore, YAP silencing decreases AR target genes’ expression, suggesting that YAP plays an important role in modulating AR transcriptional activity [146]. Additionally, higher YAP expression levels and nuclear localization positively correlate with high Gleason scores [148,149]. Consequently, YAP was identified as a prognostic biomarker for prostate cancer progression [148,149]. Considering that YAP undergoes a characteristic nuclear translocation in response to the matrix stiffening [106,107], it is conceivable that YAP activation induced by ECM stiffening could enhance AR transcriptional activity (as shown in Figure 2A). Such a mechanism could contribute to the switch from androgen-dependent to castration-resistant prostate cancer.

Altogether, this suggests the existence of a close relationship between ECM stiffness, mechanosensory proteins filamin and YAP, and AR signaling in the progression toward CRPC.

#### 3.1.3. ECM Stiffening as a Potential Driver of Androgen-Independent AR Activation

Prostate tumor progression under ADT can occur through ligand-independent AR activation since multiple growth factors and cytokines are involved in AR transactivation [150,151,152]. For instance, it has been shown that both VEGF and stromal TGF-β induce AR transactivation under androgen deprivation conditions in the LNCaP prostate cancer cell line [152,153,154]. Notably, androgen deprivation promotes an upregulation of VEGF-C, which in turn increases the AR coactivator BAG-1L expression to facilitate AR transactivation [153,154]. Moreover, TGF-β signaling induces the expression of several AR target genes, including PSA and KLK4 [152]. Among the EGFR family, HER2 tyrosine kinase modulates AR signaling activity and promotes androgen-independent prostate tumor growth in vitro and in vivo (Figure 2A) [155,156,157]. Overexpression of HER2 in the androgen-sensitive LNCaP cells allows androgen-independent expression of PSA and cell proliferation, and decreases the tumor latency of xenografts in castrated mice [155,156]. In contrast, decreased expression of HER2 by siRNA impaired LNCaP cell proliferation via targeting AR activity [157]. Together, this line of evidence presents HER2 kinase activity, as well as VEGF and TGF-β signaling, as key molecular events for optimal transcriptional activity of AR in prostate cancer cells. Furthermore, these molecular events activate downstream signaling pathways such as Ras/MAPK, PI3K/AKT, and JAK/STAT, which are known to induce cell proliferation and migration, promoting AR transactivation and the transcription of AR target genes [122,152,158,159,160]. Overall, through the positive modulation of AR transactivation, these alternate signaling pathways contribute to the development of hormone refractory tumors [160,161].

It is noteworthy to highlight that these growth factor signaling pathways are also known to be upregulated by the matrix stiffening. Notably, activation of the FAK-Src signaling network by a stiff ECM induces PI3K/AKT and MAPK/ERK signaling and contributes to tumor progression and invasion [38,116]. Alternatively, integrins can cluster with growth factor receptors, including receptors for VEGF, TGF-β, and EGF, and enhance their signaling [162,163,164,165,166,167]. For instance, formation of a HER2-Src-α6β4 integrin complex lead to an increased phosphorylation and activation of HER2, and ultimately regulates adhesion and cell proliferation [167]. This crosstalk between integrin and growth factor signaling pathways is crucial to control the cell response to matrix stiffening. Notably, in both in vivo and in vitro breast cancer models, matrix stiffening induces integrin clustering to enhance EGF-dependent ERK activation and Rho-generated force, thereby promoting tumor growth and progression [38,116,168].

In addition to the various functions of the ECM described previously, the ECM also serves as a reservoir for growth factors either found in soluble form or bound to the ECM components [169]. While some of these factors are freely available to the cells, others are not and require cell contractility to induce their release. For example, TGF-β is encapsulated within the latency-associated peptide bound to the ECM. The increased cell contractility, in response to a stiff ECM, allows cells to exert sufficient forces on the ECM-bound latency-associated peptide complex, which allows its unwrapping and subsequent TGF-β release [170]. Taken together, these data provide interesting examples on how ECM stiffness and cell contractility influence cellular responses to growth factors but also partially affect the availability of growth factors stored in the ECM.

Altogether, these examples highlight the overlap between stiffness-activated pathways and those associated with ADT resistance mechanisms. Indeed, evidence of stiffness-mediated resistance mechanisms related to the transactivation of receptor tyrosine kinases signaling pathways is available for different cancers. For instance, orthotopic injection of an estrogen receptor-positive breast tumor cell line in a genetically engineered mouse model with a fibrotic stroma showed that a stiffer environment provides increased resistance to tamoxifen [171]. In melanoma cells, increased stiffness provides resistance to BRAF inhibition by trans-activating the MAPK/ERK pathway via mechanotransduction signaling by the integrin-FAK and YAP signaling axis [172,173]. Overall, the cell response to ECM stiffening likely enhances AR activation through alternative pathways even under ADT in similar ways.

### 3.2. Non-AR-Centric Mechanisms

While many mechanisms of resistance to AR-targeted therapy are AR-centric (discussed in Section 3.1), the incidence of CRPC tumors arising through AR-independent mechanisms is increasing [7]. This is mostly due to the increased use of ARSI, leading to tumors exhibiting neuroendocrine (NE) or small-cell carcinoma (SCC) features [174], which are associated with poor clinical outcomes. Emerging evidence suggests that the WNT signaling pathway [8], as well as FGF signaling [7], are among the AR-independent mechanisms of resistance involved in progression to NE or SCC prostate cancer. Interestingly, these pathways are also related to ECM stiffness, making the case for the likely involvement of ECM stiffness in the onset of CRPC through non-AR-centric mechanisms. Noteworthy, EGFR, whose relationship with ECM (and AR) was discussed in the previous section, could also mediate the growth of metastatic prostate cancer but in an AR-independent manner [175].

#### 3.2.1. The WNT Signaling Pathway

The WNT signaling pathway is an evolutionary conserved pathway mostly known for its developmental roles, with loss-of-function experiments resulting in significant defects in organ and tissue formation [176,177]. Overall, the WNT family encompasses 19 genes in mammals that encode a plethora of cysteine-rich growth proteins [178]. These proteins bind several types of receptors and co-receptors, including the Frizzled receptors, the low-density lipoprotein receptor-related protein, receptor tyrosine kinase, and receptor tyrosine kinase-like orphan receptors [179]. The downstream signaling cascades are still being uncovered and involve canonical mediators such as β-catenin, as well as less-understood pathways such as calcium signaling.

There is mounting evidence linking the WNT signaling pathway to prostate cancer disease progression and aggressivity [180,181]. For instance, prostate cancer specimens, obtained from primary tumors as well as metastatic lesions, were found to bear mutations in the β-catenin gene [182,183]. In addition, several members of the WNT family were found to be differentially expressed in osteotropic prostate cancer cell lines (such as PC-3) compared to non-osteotropic ones (such as LNCaP and DU-145) [184], and knockdown of some of these members (e.g., *WNT5A* and *FZD*) decreases prostate cancer cell invasion [185]. Moreover, prostate cancer stem cell self-renewal abilities were found to be regulated by WNT activity, in an androgen-independent manner [186]. Additionally, β-catenin activation was found to dampen AR-mediated signaling in murine prostate cancer models and altered WNT pathway genes were associated with a decreased overall survival in mCRPC patients [8]. Altogether, there is emerging evidence linking prostate cancer progression to alterations in the WNT signaling pathway.

Interestingly, the WNT pathway is responsive to increased ECM stiffness. In fact, many components of the WNT signaling pathway, such as WNT1, WNT3A, and β-catenin, are upregulated by stiffer matrices (Figure 2B), which was confirmed both at the transcript and the protein levels, and found to be initiated mainly through integrin/FAK signaling, in bone marrow stem cell and primary chondrocyte models [187]. Moreover, MMP-7 was demonstrated to be activated by a stiff matrix in a colorectal cancer model in vitro [188]. Interestingly, this matrix-regulating protein was also found to crosstalk with the WNT pathway to drive prostate cancer progression [189].

Despite the lack of a current understanding directly linking ECM stiffening to the WNT pathway in the context of prostate cancer, it is tempting to speculate that the effects of the ECM on the WNT pathway in other physiological and pathological processes could also be relevant to prostate cancer, where stiffer matrices can trigger WNT signaling, which subsequently impacts prostate cancer progression and development (Figure 2B).

#### 3.2.2. The FGF Signaling Pathway

The FGF family groups a large number of ligands that interact with four different tyrosine kinase receptors, the FGF receptors or FGFRs, regulating key cellular functions such as cell division, migration, and differentiation [190]. The functions they assume can be autocrine, paracrine, or endocrine. Disease-causing mutations in this pathway are well-documented, spanning a wide range of organs and systems such as the heart, the lung, the skin, the brain, and the urinary system [190]. The pathway’s role in cancer development in general as well as its therapeutic potential have been reviewed elsewhere [191].

The FGF/FGFR signaling axis has been shown to be required for the development of the prostate gland [192], as well as the maintenance of the adult gland homeostasis, where human prostatic cells produce and secrete FGFs and express FGFRs, making them responsive to autocrine and paracrine signals that support normal gland growth, irrespective of androgen levels [193]. Deregulation in this pathway gives rise to life-threatening neoplasms, with several members of the FGF family (e.g., FGF1, FGF2, FGF6, and FGF8) found to be upregulated in prostate cancer models and in clinical settings [194]. Conjunctively, constitutive activation of certain FGFRs (such as FGFR1) usually accompanies prostatic intraepithelial neoplasia onset, setting the stage for prostate cancer development [195].

In the context of AR-independent prostate cancer, which usually emerges as a result of ARSI therapies, FGF signaling seems to play a pivotal role, where non-neuroendocrine AR-negative tumors rely on FGF and MAPK signaling to bypass AR requirements in prostate cancer (Figure 2B) [7]. In addition, human AR-negative prostate cancer patient-derived xenografts were found to exhibit strong osteoinductive activity, all while overexpressing FGF9 and WNT ligands, highlighting the role of both pathways in mediating AR-independent prostate cancer progression and metastatic spreading to the bone [196].

Unlike the WNT pathway, few studies have investigated the potential link between FGF signaling and ECM dynamics. One developmental study found that mechanical stress, which can be increased due to ECM stiffness [197], activates the FGFR/ERK2 pathway during embryogenesis [198]. It has long been argued that cancer is a “problem of developmental biology” [199], with several developmental processes being reactivated during tumorigenesis [200], including FGF signaling [191]. Moreover, FGFs are stored within the ECM and can affect disease course (Figure 2B) [201]. In addition, ECM-free FGF2 was found to be increased in thyroid carcinomas [202]. In breast cancer, many components and cellular mediators of the ECM were found to be tightly linked to the FGF axis, impacting tumor cell response and hormone dependence. For instance, cancer-associated fibroblasts were found to promote hormone independence by upregulating FGF2 in a murine breast cancer model [203]. In addition, tumor-associated macrophages were shown to be recruited by FGFR1 activation in mammary epithelial cells [204], further underlining the importance of this axis in tumorigenesis. Intriguingly, obesity-induced adipocyte diameter increase was correlated with FGF and FGFR1 activation, and progression towards breast cancer hormone independence [205]. Since breast and prostate cancers share key similarities regarding hormone dependency and associated hormonal therapy resistance [206], it is plausible that similar mechanisms dictating hormone independence and involving stimulation of cancer-associated fibroblasts and macrophages by members of the FGF family and are at play in prostate cancer as well. In this scenario, an increase in FGF signaling could recruit tissue-resident macrophages and cancer-associated fibroblasts to the tumor site, promoting ECM remodeling and ultimately affecting the course of tumor progression.

## 4. Conclusions

In conclusion, the ECM remains an important piece of the cancer puzzle, as knowledge advances beyond cell-centric features and environmental factors gain more traction and better understanding. This is being reflected in clinical trials, where drugs specifically targeting the ECM are being developed and tested in several cancers [9,64,207,208]. Prostate cancer is a special case due to its hormonal origins and extremely slow-growing nature, which increases the challenges associated with studying this disease. Therefore, the current state of the literature regarding ECM stiffness in prostate cancer disease is incomplete. To address this gap, we looked at other cancer types for clues on how the ECM can promote cancer progression and highlighted signaling pathways that are known to be activated during prostate cancer progression and by increased ECM stiffness. However, more research is needed to better characterize the relationship between the mechanical aspect of matrix stiffness and the molecular features of prostate cancer, especially with regards to AR signaling. The recent advances made in mechanobiology and live microscopy will certainly enable a deeper understanding of how prostate cancer cells interact with their environment in the upcoming years.

If ECM stiffening is effectively involved in CRPC through both AR-dependent and AR-independent mechanisms, targeting either the ECM stiffening or the tumor cell mechanosensing abilities is tempting. Similar therapeutic strategies are in fact actively explored in other cancers [207]. The clinical relevance of such strategy is huge since targeting both AR-positive and AR-negative prostate cancer cell populations would provide a two-pronged approach to eliminate CRPC cells, irrespectively of AR status.

## Figures and Tables

**Figure 1 cancers-14-02887-f001:**
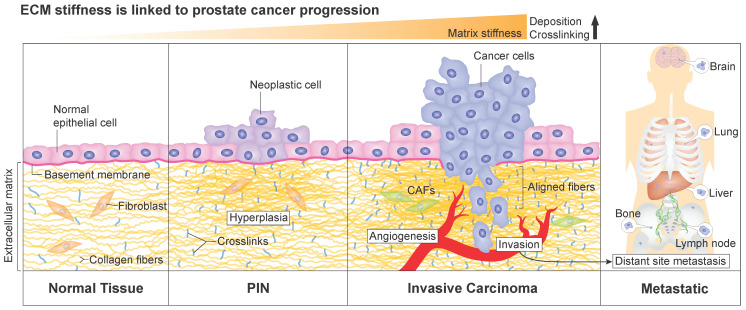
Gradual increase of ECM stiffness and prostate cancer progression. As prostate cells slowly progress towards prostatic intraepithelial neoplasia (PIN) and ultimately into invasive disease, the extracellular matrix that supports normal tissue homeostasis is gradually altered, which results in a stiffer prostate tissue. The matrix stiffness increases over time due to a greater rate of matrix components’ deposition (including type I collagen, pictured here), formation of crosslinking between ECM fibers, and ECM remodeling mediated mainly by fibroblasts and cancer-associated fibroblasts (CAF). These ECM alterations ultimately influence the hallmarks of cancer, especially angiogenesis, invasion, and distant site metastasis. Notably, fibers’ alignment can serve as paths that guide cancer cell migration.

**Figure 2 cancers-14-02887-f002:**
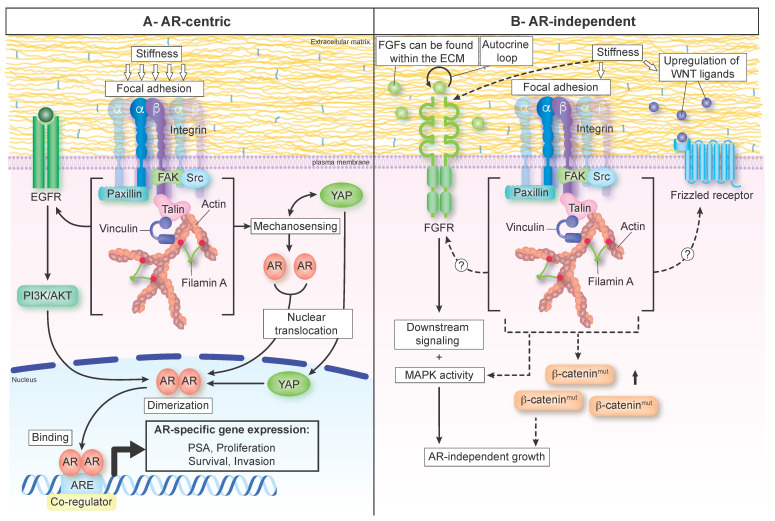
Signaling pathways activated by ECM stiffness and linked to castration resistance mechanisms in prostate cancer. (**A**) AR-centric mechanisms. Matrix stiffness, through an excessive ECM protein deposition and their crosslink, is sensed by focal adhesions, which can in turn activate EGFR or promote YAP nuclear translocation, a mediator of the HIPPO pathway. Activated PI3K/AKT downstream of EGFR could then promote AR translocation to the nucleus, followed by its dimerization, allowing it to bind to specific regions in the DNA to trigger pro-tumorigenic cellular responses (proliferation, invasion, and survival). Nuclear YAP could also bind AR and enhance its transcriptional activity. (**B**) AR-independent mechanisms. Matrix stiffness sensed by focal adhesions can potentially activate FGFR or modulate the WNT pathway to drive AR-independent growth of prostate cancer. FGFs that are stored in the ECM can serve as ligands of the FGF signaling axis, and stiffness-activated FGFR will turn on downstream signaling (MAPK) to drive prostate cancer progression. Alternatively, the WNT pathway’s activation by increased stiffness can result from upregulation of the WNT ligands or increase sensitivity of the Frizzled receptor to its ligands. Increase in mutant β-catenin levels could further facilitate the crosstalk between focal adhesions and the WNT pathway. Altogether, these mechanisms could contribute to ECM stiffness-driven prostate cancer disease progression. AR: androgen receptor; ARE: androgen response element; EGFR: epidermal growth factor receptor; FAK: focal adhesion kinase; FGFR: fibroblast growth factor receptor; FGFs: fibroblast growth factors; PSA: prostate-specific antigen; YAP: Yes-associated protein.

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
