# Peer review of "The Extracellular Matrix Stiffening: A Trigger of Prostate Cancer Progression and Castration Resistance?"

_cancers, 2022, doi:10.3390/cancers14122887_

Round 1

Reviewer 1 Report

A well presented literature review article on the role of ECM stiffness and prostate cancer progression highlighting some key areas of focus and the complexities in developing therapeutic targeting.

While discussions of the ECM were well presented a key area of focus being the proteoglycan family particularly versican and it's regulatory partners the metzincin super family seem to have been omitted and would be a strong addition in discussion of ECM regulation and stiffness. The reviewer notes there is mention of the MMP family however this could be further expanded to other family members.

There also seems to be some overlap in sections which could be further condensed for example, section 4 could be combined with section 2.3 and allow for further detailed discussion on the ECM.

Figure 1 could have an addition after invasion to include metastasis as there are key regulations going on between invasion and metastasis in prostate cancer progression.

Figure 2 is it the increased matrix stiffness which leads to these events? Both A and B look the same level of stiffness is this correct? There also appears to be crosslinks missing as described in your first figure.

Spelling/grammar 

-Line 46 - because the AR is activated?

Reviewer 2 Report

Very good review, highlighting an interesting and important topic, which could be very useful to people in the cancer field.

Round 2

Reviewer 1 Report

Dear authors,

Well done on making the changes suggested they all seem to be satisfactorily addressed apart from figure 2 are there still cross links missing in the revised script? Have a look at your figure 1 and make sure these are replicated.

The reviewer also suggests to consider revision of lines 228-232 which are not clear if you are talking about colon cancer or prostate.

Line 248 space required between word and reference

Author Response

Please see the enclosed document.
